# Vector Quantization in the Brain: Grid-like Codes in World Models

**Xiangyuan Peng**[*]
1900012762peng@pku.edu.cn

**Xingsi Dong**[*†]
dxs19980605@pku.edu.cn

**Si Wu**[†]
siwu@pku.edu.cn

PKU-Tsinghua Center for Life Sciences, Academy for Advanced Interdisciplinary Studies.
IDG/McGovern Institute for Brain Research, Center of Quantitative Biology, Peking University.
School of Psychological and Cognitive Sciences,
Key Laboratory of Machine Perception (Ministry of Education).
[*]: Equal contribution.
[†]: Corresponding authors.

## Abstract

We propose Grid-like Code Quantization (GCQ), a brain-inspired method for compressing observation–action sequences into discrete representations using grid-like patterns in attractor dynamics. Unlike conventional vector quantization approaches that operate on static inputs, GCQ performs spatiotemporal compression through an action-conditioned codebook, where codewords are derived from continuous attractor neural networks and dynamically selected based on actions. This enables GCQ to jointly compress space and time, serving as a unified world model. The resulting representation supports long-horizon prediction, goal-directed planning, and inverse modeling. Experiments across diverse tasks demonstrate GCQ's effectiveness in compact encoding and downstream performance. Our work offers both a computational tool for efficient sequence modeling and a theoretical perspective on the formation of grid-like codes in neural systems.

## 1 Introduction

VQ-VAE [1] introduces discrete latent variables into the autoencoding framework through vector quantization (VQ) [2], allowing the model to compress high-dimensional continuous inputs into discrete, tokenized representations. This capability has led to its widespread application across various domains, including images [3, 4], video [5], speech [6], actions [7], and multimodal data [8], demonstrating its versatility in handling complex, diverse inputs. The success of VQ-VAE underscores the utility of compressing inputs into reusable codes as a general computational strategy for preprocessing and organizing data across a wide range of tasks.

Biological systems face the similar challenge: how to process and represent high-dimensional, continuous inputs arising from multiple sensory and motor modalities. In parallel, the brain exhibits grid-like codes (GCs), which serve as general-purpose neural patterns for encoding information. GCs are extensively observed across various brain regions. Initially identified in the medial entorhinal cortex for spatial navigation [9], GCs have since been observed in the neocortex [10, 11, 12] and associated with representing abstract concepts beyond space, such as time and relational knowledge [10, 11, 13, 14]. This widespread neural activity is characterized by bump-like patterns, periodicity, and typically disentangled representations.

Building on this insight, we propose a brain-inspired VQ method, Grid-like Code Quantization (GCQ), which uses the principles of GCs to structure the codebook. Specifically, we use continuous attractor neural networks (CANNs) [15, 16, 17] to generate grid-like activity patterns, where each stable

39th Conference on Neural Information Processing Systems (NeurIPS 2025).

state—bump—acts as a codeword. Due to the finite number of neurons, these bumps naturally form a discretized representation [18]. Unlike traditional VQ methods that use a static codebook, GCQ introduces an action-conditioned codebook: a dynamic set of codewords formed by CANN-generated bumps whose transitions are modulated by actions. This enables GCQ to perform quantization not on isolated observations, but on observation–action sequences, allowing the representations to capture temporal dependencies and behavioral context. Moreover, assigning distinct CANNs to different action types naturally yields disentangled representations, facilitating generalization and compositionality.

The overall GCQ pipeline follows an encoder–quantizer–decoder architecture, adapted for action-conditioned sequence compression (Fig. 2). Specifically, the model processes an observation–action sequence, where the action sequence is used to construct an action-conditioned codebook, and the observation sequence is passed through the encoder to produce a corresponding latent sequence. This latent sequence is then quantized via template matching with the action-conditioned codebook. The matched codewords are passed to the decoder, which reconstructs the original observation sequence. Since the codebook is fixed, training requires only a commitment loss and a reconstruction loss. To enable gradient flow through the discrete quantization step, we use a straight-through estimator (STE).

GCQ is a dynamic compression approach that operates on observation–action sequences, and therefore serves as a form of world model [19, 20]. Unlike prior world models that rely on a two-stage design to separately compress space and time—typically using models like VQ-VAE for static spatial observations and autoregressive models [21] for temporal dynamics—GCQ performs spatial and temporal compression jointly.

In summary, our contributions are as follows:

- To the best of our knowledge, GCQ is the first model to unify spatial and temporal compression through an action-conditioned quantization process. This enables direct compression of observation–action sequences, offering an integrated alternative to conventional two-stage world models. (Sec. 4)

- GCQ's spatiotemporal compression yields a cognitive map, which supports long-horizon prediction, goal-directed planning, and the derivation of an inverse model. In particular, goal-directed planning becomes computationally simple, as it reduces to finding a sequence of valid bump transitions on the map. (Sec. 5).

- GCQ offers insights into the formation of GCs in the brain, enhancing our understanding of neural representations (Sec. 6).

## 2  Related Work

**VQ methods** Vanilla VAEs [22] often suffer from posterior collapse in their latent spaces when compressing high-dimensional data [23], impairing downstream tasks. VQ-VAEs [1] address this by enforcing a structured latent space through discretization. Due to their superior compression efficiency and tokenization paradigm, VQ has become a standardized module in single-modal preprocessing pipelines in machine learning [3, 4]. In multimodal settings, these compressed tokens further act as a universal interface across modalities [5]. Meanwhile, numerous studies have proposed diverse codebook designs to enhance compression rates [24, 25, 26]. Unlike most learnable codebooks, FSQ uses a predefined codebook. Similarly, our GCQ utilizes a fixed codebook derived from continuous attractor dynamics. Critically, our method diverges from conventional VQ approaches by performing sequence-to-sequence template matching rather than single-frame matching.

**World models** [19, 20] provide a framework for predicting future observations conditioned on actions. Most world models based on encoder–decoder architectures first compress observations using a VAE, and then model temporal dynamics in the latent space using temporal predictors such as RNNs [27, 28], Transformers [29], S4 models [30], or continuous Hopfield networks [31]. These approaches typically follow a two-stage design, with spatial and temporal compression handled separately. In contrast, GCQ is also an encoder–decoder world model, but it performs spatial and temporal compression jointly. There also exist decoder-only world models [32] that skip explicit compression and directly predict future observations. However, these models often struggle with planning due to the high computational cost of operating in the raw observation space. GCQ, by

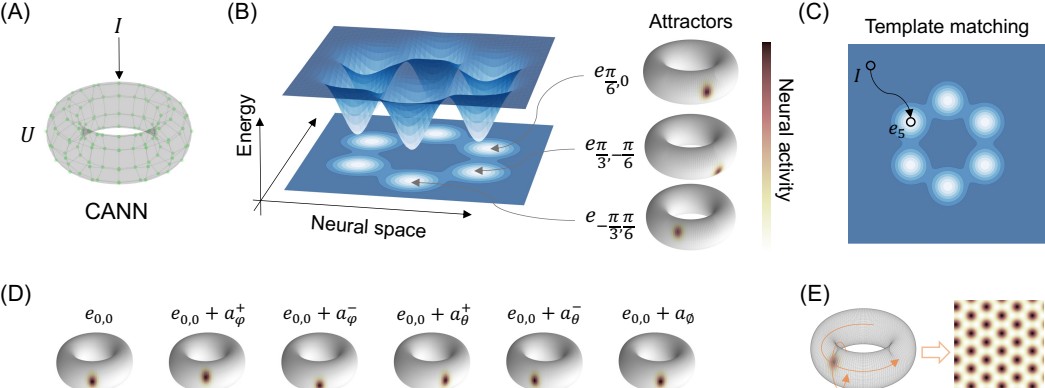

Figure 1: (A) Schematic of a CANN: Each green dot represents a neuron uniformly distributed on a torus. The neurons receive external input $I$. (B) Energy landscape of CANN dynamics: Each local minimum in the energy landscape corresponds to an attractor state, which manifests as a 2D Gaussian bump on the torus. (C) Template matching via CANN dynamics: The CANN inherently performs template matching between the external input $I$ and its attractor states. The input $I$ is matched to the attractor that maximizes their inner product. (D) Attractor transition: Under four distinct actions, the attractor initially at position $(0,0)$ stabilizes to four new attractor states. (E) Due to the periodic boundary conditions of the CANN, bump movements along the two axes naturally form grid-like patterns.

compressing both space and time into a compact latent representation, enables more efficient planning and inference.

**Cognitive map with CANNs** Unlike classical attractor networks [33]—which store discrete, unstructured patterns—CANNs encode structured patterns organized by metric relationships. This geometric regularity facilitates flexible state transitions through predefined operators [34], enabling operations like metric-based navigation and relational inference. Recent advances [35] have harnessed predefined CANNs as structured latent states for representation learning, empirically validating their ability to model neural population dynamics. Further work [36] proposes that structured latent spaces can map biologically to the entorhinal-hippocampal loop, a core circuit for spatial and episodic memory. However, existing implementations rely on biologically constrained online learning, which limits scalability. Our GCQ framework uses offline learning, enhancing parallelism and enabling application to large-scale datasets.

## 3   CANNs and Template Matching

In this section, we will briefly introduce CANNs, explain how they can form bumps as attractor states. In parallel, for VQ, the latent state obtained by the encoder must undergo template matching with codewords. We will demonstrate that CANNs inherently implement template matching between representations and bump states through their intrinsic dynamics. Finally, we will show how transitions between distinct attractor states can be mediated by actions.

GCs can naturally be modeled by bumps in CANNs (Fig. 1E). The formation of CANNs does not require complex optimization but relies on translation-invariant connectivity and periodic boundary conditions. CANNs have been widely used as canonical models to elucidate the encoding of features in neural systems, including, for example, the encoding of orientation [37], head direction [38] and spatial location [17, 39]. CANNs can be expressed through various mathematical formulations. Here, we adopt a relatively concise form [16] to demonstrate their principles. We consider $N^2$ neurons distributed on a toroidal ($S^1 \times S^1$) surface. These neurons are indexed by their positions on the torus $\theta \in \{\theta_i\}_{i=1}^N$ and $\varphi \in \{\varphi_j\}_{j=1}^N$, where $\theta_i$ and $\varphi_j$ are uniformly distributed over $(-\pi, \pi]$ (Fig. 1A). Let $U_{\theta,\varphi}(t)$ and $r_{\theta,\varphi}(t)$ denote the synaptic input and firing rate, respectively, of the neuron located

at $(\theta, \varphi)$ at time $t$. The dynamics of the CANN are governed by:

$$\tau \frac{\partial U_{\theta,\varphi}(t)}{\partial t} = -U_{\theta,\varphi}(t) + \rho \sum_{\theta',\varphi'} W_{\theta,\varphi}(\theta', \varphi') r_{\theta',\varphi'}(t) + I_{\theta,\varphi}(t), \quad (1)$$

where $\tau$ is the synaptic time constant and $\rho$ is the neuronal density. $W_{\theta,\varphi}(\theta', \varphi')$ is the recurrent neuronal connections weights between neuron $(\theta, \varphi)$ and neuron $(\theta', \varphi')$,

$$W_{\theta,\varphi}(\theta', \varphi') = \frac{J}{2\pi a^2} \exp\left[-\frac{\|\theta - \theta'\|_S^2 + \|\varphi - \varphi'\|_S^2}{2a^2}\right]. \quad (2)$$

The norm $\|\cdot\|_S$ denotes the shortest path between two points on the circle, ensuring periodic boundary and translation-invariant conditions. The parameters $J$ and $a$ control the strength and width of the Gaussian connectivity, respectively. The nonlinear relationship between the firing rate $r_{\theta,\varphi}(t)$ and the synaptic input $U_{\theta,\varphi}(t)$ is implemented by divisive normalization, which is written as,

$$r_{\theta,\varphi}(t) = \frac{U_{\theta,\varphi}^2(t)}{1 + k\rho \sum_{\theta',\varphi'} W_{\theta,\varphi}(\theta', \varphi') U_{\theta',\varphi'}^2(t)}, \quad (3)$$

where $k$ controls the normalization strength. In reality, divisive normalization could be implemented by shunting inhibition [40].

Previous studies [16, 41] have established that the CANN dynamics governed by Eq. (1) possess $N^2$ stationary states (attractors) when the external input $I_{\theta,\varphi}(t) = 0$ (Fig. 1B). Each state corresponds to a 2D Gaussian bump on the torus, centered at coordinates $(\theta, \varphi)$, with the firing rate of the neuron at position $(\theta', \varphi')$ given by:

$$e_{\theta,\varphi}(\theta', \varphi') = A \exp\left[-\frac{\|\theta - \theta'\|_S^2 + \|\varphi - \varphi'\|_S^2}{2a^2}\right], \quad (4)$$

where $A = \left[1 + (1 - 32\pi a^2 k/J^2\rho)^{1/2}\right]/(4\pi a^2 k\rho)$ is the amplitude. When $I_{\theta,\varphi}(t)$ is a constant input, prior work [42] demonstrated that after its removal, the network converges to an attractor determined by:

$$\theta^*, \varphi^* = \max_{\theta,\varphi} \sum_{\theta',\varphi'} e_{\theta,\varphi}(\theta', \varphi') I_{\theta',\varphi'}, \quad (5)$$

which demonstrates that CANN dynamics effectively perform template matching between the input $I$ and the $N^2$ attractors according to their inner product. (Fig. 1C).

The bump in CANNs exhibit high mobility, enabling controlled movement through mechanisms such as: anti-symmetric connections [43], negative feedback [44, 43], velocity neurons [45]. Such bump displacements correspond to transitions between attractor states. For the toroidal CANN described above, each attractor can undergo local two-dimensional displacements in the $\theta, \varphi$ plane. We define two orthogonal action bases aligned with the $\theta$ and $\varphi$ axes (Fig. 1D),

$$a_\theta^\pm = e_{\theta\pm\Delta\theta,\varphi} - e_{\theta,\varphi}, \quad a_\varphi^\pm = e_{\theta,\varphi\pm\Delta\varphi} - e_{\theta,\varphi}. \quad (6)$$

where $\Delta\varphi$ and $\Delta\theta$ denote a small displacement step.

## 4 Grid-like Code Quantization

In this section, we first introduce the action-conditioned codebook in GCQ and the template matching process for sequences. We then describe how GCQ enables bidirectional mapping between real-world actions and latent transitions, and propose a greedy operator for measuring distances on the cognitive map to support inverse modeling and planning.

### 4.1 Action-conditioned codebook and sequence matching

We first introduce the key difference between GCQ and VQ from a high-level perspective. In the VQ method, the encoder first compresses the observation $o$ into $s$, which is then matched to the closest codes in the codebook through template matching, producing $\hat{s}$. The decoder then reconstructs $\hat{o}$ from $\hat{s}$. In GCQ, the input consists of an action-observation sequence $\{o_1, a_1, o_2, a_2, ..., o_n\}$. The encoder

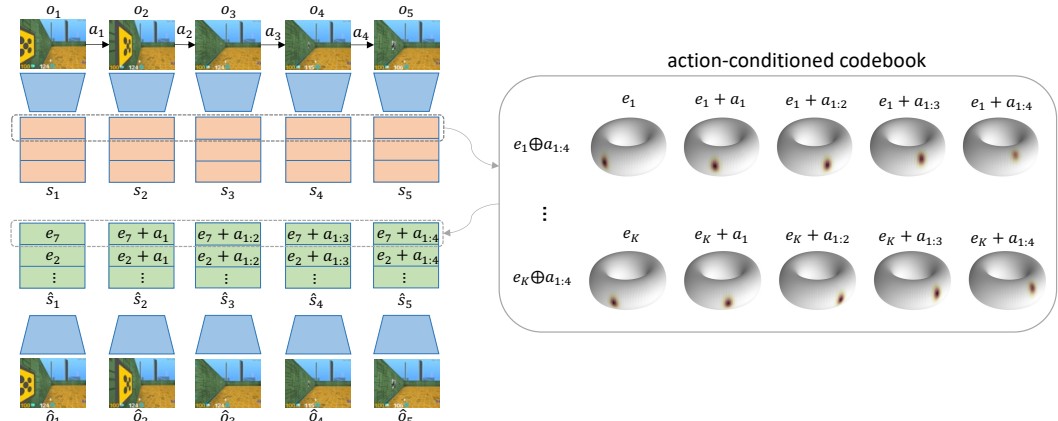

Figure 2: Schematic of GCQ: The action-observation sequence is encoded by the encoder into a latent state composed of $m = 3$ codes. Through sequence template matching with the action-conditioned codebook, the decoder reconstructs the predicted observations. The gray arrows indicate the template matching process, and the gray dashed boxes represent the matching targets.

compresses the observation sequence $o_{1:n} = \{o_1, o_2, ..., o_n\}$ into $s_{1:n}$, which is then matched to the closest codes in the action-conditioned codebook via template matching, yielding $\hat{s}_{1:n}$. The decoder then reconstructs $\hat{o}_{1:n}$ from $\hat{s}_{1:n}$.

In GCQ, each code corresponds to an attractor in the CANN (Fig. 1B). In the previous section, we used $\theta$ and $\varphi$ to index different attractors; for simplicity, we will now use natural numbers as attractor indices. Each code consists of $d$ neurons, and the codebook contains $K$ attractors. A simple implementation sets $K = d$, where each attractor's center coincides with a single neuron. Alternatively, we can set $K > d$, causing some attractor centers to fall between two neurons. In practice, different combinations of $K$ and $d$ can be selected. The state representation $s_i \in \mathbb{R}^{m \times d}$, meaning that $s_i$ is composed of $m$ codes.

Additionally, we manually define a mapping between the action sequence $a_i \in \mathcal{A}$ from the dataset and the action combinations applied to the CANNs. For notational simplicity, we hereafter use $a_i$ to refer to an action in either the original space or the CANN space. In the latter context, $a_i \in \mathbb{R}^{m \times d}$ represents the composite action over $m$ bumps, with its component $a_i^j \in \mathbb{R}^d$ denoting the action applied to the $j$-th bump in Eq.(6). Each CANN supports five distinct actions, resulting in up to $5^m$ possible action combinations across $m$ CANNs. Since this mapping is injective, the discrete action space must satisfy $|\mathcal{A}| \leq 5^m$. For continuous actions, a CANN can define transitions in two directions, imposing the constraint $\dim(\mathcal{A}) \leq 2m$.

After establishing the mapping, we quantize the latent representation $s_{1:n} = \{s_{1:n}^j\}_{j=1}^m$. This representation consists of a set of $m$ parallel sequences, where each $s_{1:n}^j$ corresponds to a sequence from one of the $m$ CANNs (as depicted by the dashed lines in Fig. 2). The quantization process is performed independently for each of these $m$ sequences. For each latent sequence $s_{1:n}^j$, we perform a template matching procedure. This involves comparing $s_{1:n}^j$ against a set of $K$ candidate trajectories. Each candidate trajectory is generated by applying the known action sequence $a_{1:n-1}^j$ to a base bump state $e_i$. We denote this operation as:

$$e_i \oplus a_{1:n-1}^j = \{e_i, e_i + a_1^j, \ldots, e_i + a_{1:n-1}^j\}, \tag{7}$$

where $e_i + a_{1:n-1}^j = e_i + \sum_{t=1}^{n-1} a_t^j$. The index $k$ of the best-matching codeword for the $j$-th latent sequence is found by minimizing a distance metric (e.g., the L2 norm) between the latent sequence and each of the $K$ candidate trajectories:

$$k_j = \arg\min_{i \in \{1,..,K\}} ||s_{1:n}^j - (e_i \oplus a_{1:n-1}^j)|| \tag{8}$$

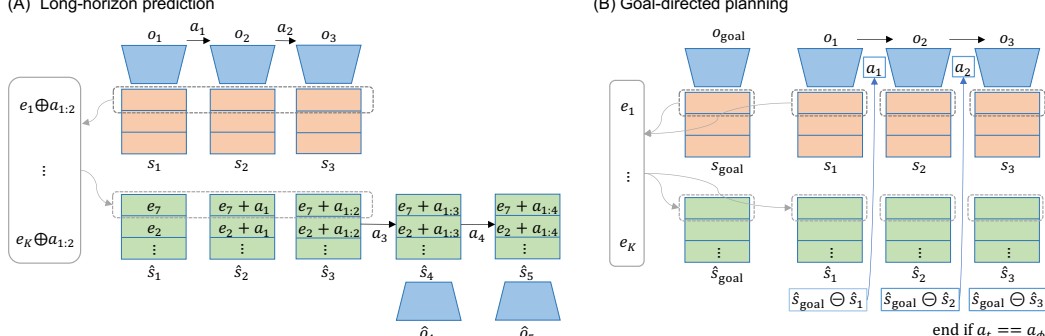

Figure 3: (A) Schematic of long-horizon prediction. The figure illustrates the process of initializing with a sequence of length 3 and predicting two future observations. (B) Schematic of goal-directed planning. Some gray arrows are omitted for clarity. The blue arrows represent the mapping from actions in the latent space to real agent actions. Through iterative action generation, environment interaction, and observation, the agent continues until it outputs a no-op action $a_\emptyset$, indicating that the goal has been reached.

Finally, the quantized sequence $\hat{s}_{1:n}^j$ is constructed using this optimal codeword $e_{k_j}$. The complete quantized representation $\hat{s}_{1:n}$ is the collection of these individually quantized sequences:

$$\hat{s}_{1:n}^j = e_{k_j} \oplus a_{1:n-1}^j, \quad \text{and} \quad \hat{s}_{1:n} = \{\hat{s}_{1:n}^j\}_{j=1}^m \tag{9}$$

When computing the loss in GCQ using backpropagation (BP), we adopt the same straight-through estimator (STE) as in the VQ method, copying gradients from the decoder input to the encoder output to enable gradient flow to the encoder. GCQ uses two loss terms: a reconstruction loss and a commitment loss:

$$L = ||o_{1:n} - \hat{o}_{1:n}||^2 + \beta \, ||s_{1:n} - \text{sg}\,[\hat{s}_{1:n}]||^2 \tag{10}$$

where sg[·] denotes the stop-gradient operation and $\beta$ adjusts the strength of the commitment loss.

In GCQ, the encoder and decoder are not designed in the same way as in conventional VQ models. Traditional VQ architectures often use ResNet-based building blocks, which provide each code with only a limited receptive field. As a result, modifying a single code typically leads to only local changes in the reconstructed observation. In contrast, GCQ assigns each code to an action, and altering the action can result in global changes to the observation. This necessitates that each code has access to global information during encoding and decoding. To address this, we explore three architectural variants for the encoder and decoder: (1) ResNet followed by a fully connected layer, (2) ViT [46], and (3) a hybrid of ResNet and ViT. Among these, ViT achieves the best trade-off in terms of parameter efficiency, training stability, and overall performance (Table.1).

## 4.2 Operations on cognitive map

GCQ uses a structured latent space, allowing an agent's actions in the real environment to correspond to simple movements of bumps within the latent space. In effect, GCQ constructs a space defined by bump dynamics, which can be interpreted as a cognitive map. By establishing a mapping between observations and this map, actions in the real space can be projected onto the map to determine position changes, and conversely, movements within the map can be mapped back to real-space actions. This bidirectional mapping enables GCQ to support both inverse modeling and goal-directed planning. Specifically, to compute the distance between two states $s_i$ and $s_j$, we define an operation on the cognitive map. Since bump movements are action-driven and only valid actions produce feasible transitions, we introduce the following operation:

$$s_i \ominus s_j = \arg\min_{a \in \mathcal{A}} |s_j + a - s_i|. \tag{11}$$

This operation represents a greedy step: it selects the best valid action $a$ that moves $s_j$ one step closer to $s_i$.

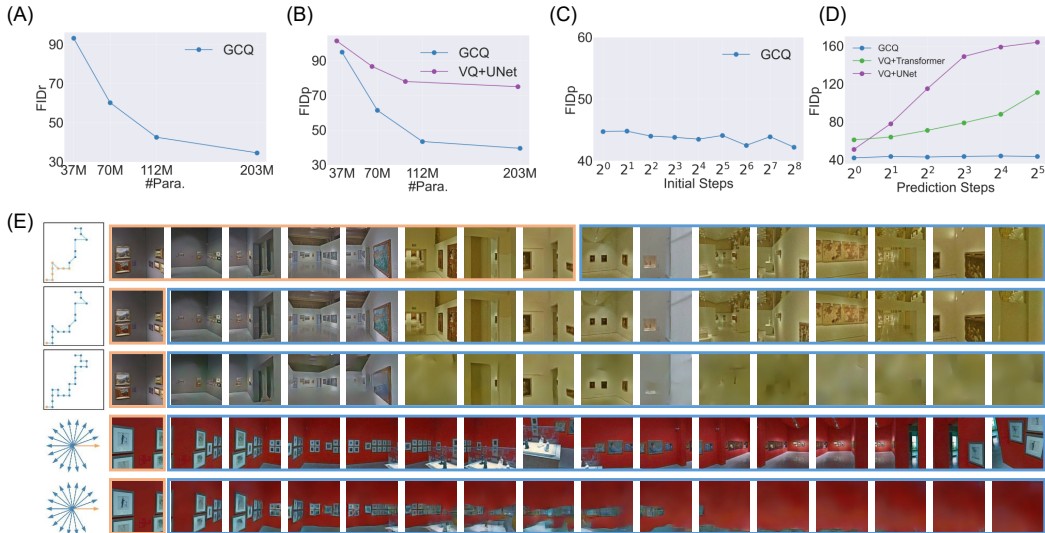

Figure 4: (A)(B) Reconstruction FID and prediction FID for GCQ and VQ+UNet across different model sizes. (C) Prediction FID of GCQ varies with changes in the initialization length. (D) Prediction FID of GCQ (#Para:112M), VQ+UNet (96M) and VQ+Transformer (121M) changes as the prediction length increases. (E) Predictions on GSV dataset. The first patch in each row represents the trajectory drawn by the action. The first three rows correspond to movement actions, while the last two rows correspond to rotation actions. In practice, different actions are encoded within a single GCQ, enabling their use. For visualization convenience, they are plotted separately here. Rows 1, 2, and 4 show GCQ predictions with different initialization lengths (orange frames), predicting subsequent observations (blue frames). Rows 3 and 5 show VQ+UNet predictions under the same conditions. As the prediction length increases, the images become blurry.

## 5 Experiment

As a spatiotemporal compression model, GCQ is first evaluated in ablation studies to demonstrate its ability to compress and reconstruct observations. We then show that GCQ, when used as a world model, supports long-horizon prediction, goal-directed planning, and inverse modeling. Compared to traditional two-stage models, GCQ exhibits superior performance in long-range prediction tasks.

**Datasets.** We evaluate GCQ on four datasets, all of which contain image-based observations. The 2DMaze [47] dataset is a virtual environment where actions correspond to the agent's movements. Each observation contains a full view of the maze, providing complete information. The Google Street View (GSV) dataset represents real-world environments with partial observations; the actions include both translational movements and rotational head turns in two directions. In the MPI3D [48] and 3DShapes [49] datasets, actions are defined as abstract feature-level changes.

**Baselines.** We compare GCQ with traditional two-stage world models. VQ-VAE is used in the first stage for spatial compression. The codebook size in GCQ and VQ-VAE is kept the same for a fair comparison. For modeling temporal relationships, we use a UNet that predicts the next latent state $s_{t+1}$ based on the current latent state $s_t$ and action $a_t$. We refer to this baseline as 'VQ+UNet.' For action embedding in the UNet, we follow the approach from LAPO [7]. To further model temporal dependencies, we also adopt a Transformer-based architecture following TransDreamer [29]. We refer to this baseline as 'VQ+Transformer.'

**Evaluation Metrics.** To evaluate the quality of the model-generated observations, we report peak signal-to-noise ratio (PSNR) for pixel-level reconstruction fidelity, and use the Fréchet Inception Distance (FID) [50] to assess the quality of generated images.

**Ablations** We first conducted ablation experiments on the GSV dataset. Table 1 presents the performance of three different encoder-decoder network building blocks. It can be observed that the ViT and Hybrid models achieve better performance with fewer parameters. However, during the

experiments, we found that the Hybrid model was less stable in training and converged more slowly than ViT. Therefore, unless otherwise specified, all subsequent experiments utilized the ViT-structured network. The GCQ exhibits scalability with model size similar to VQ+UNet, both in reconstruction and prediction. (Fig. 4A,B).

| Model type | Image size | #Para. | FIDr↓ | FIDp↓ | PSNRr↑ | PSNRp↑ |
|---|---|---|---|---|---|---|
| Resnet | $3 \times 80 \times 40$ | 330M | 48.05 | 48.57 | 25.70 | 25.59 |
| Hybrid | $3 \times 80 \times 40$ | 64M | 21.29 | 22.31 | 29.07 | 28.54 |
| | $3 \times 128 \times 128$ | 140M | 41.55 | 41.91 | 26.31 | 25.59 |
| ViT | $3 \times 80 \times 40$ | 90M | 13.27 | 13.92 | 31.32 | 31.34 |
| | $3 \times 128 \times 128$ | 112M | 42.56 | 43.41 | 27.82 | 27.77 |

Table 1: Model comparisons on the Street View dataset. FIDr, FIDp, PSNRr, and PSNRp represent the FID and PSNR scores for reconstruction and prediction, respectively. ↓ and ↑ indicate that lower or higher values are better. The ResNet model was not trained on higher resolution images because its architecture includes a fully connected layer after the convolutional backbone, resulting in an excessively large number of parameters.

We also make the bump-like codes in the codebook learnable by using the following loss function:

$$L = ||o_{1:n} - \hat{o}_{1:n}||^2 + \beta \left\| s_{1:n} - \text{sg}\left[\hat{s}_{1:n}\right]\right\|^2 + \gamma \left\| \text{sg}\left[s_{1:n}\right] - \hat{s}_{1:n}\right\|^2 \tag{12}$$

However, our experiments show that making the codes learnable actually degrades performance. We attribute this to the fact that, unlike the relatively simple codes in VQ, our codes exhibit more complex dynamic relationships. Allowing the codes themselves to be trained may therefore reduce training stability.

| Model | FIDp↓ | PSNRp↑ |
|---|---|---|
| GCQ (fixed) | 43.41 | 27.77 |
| GCQ (learnable) | 47.76 | 24.48 |

Table 2: Effect of learnable vs. fixed codes in GCQ.

**Long-horizon prediction.** After being initialized with an observation-action sequence, GCQ can perform actions directly in the latent space to predict future observations (Fig. 3A). Notably, its prediction performance remains stable regardless of the length of the initialization sequence (Fig. 4C; Fig. 4E, rows 1–2). As shown in Fig. 4D, the performance of the VQ method degrades as the prediction horizon increases, whereas GCQ maintains robust predictive quality due to its stable latent structure (Fig. 4E, rows 2–3). This is a key advantage of GCQ: by constructing a consistent cognitive map, it effectively addresses the instability issues commonly seen in current world models [51]—such as inaccurate predictions after completing a full rotation in the environment (Fig. 4E, rows 4–5). GCQ also demonstrates strong zero-shot prediction capabilities on relatively simple datasets. As shown in Fig. 5, rows 1–2, the model produces reasonable predictions in environments it has never encountered during training. Furthermore, by treating abstract feature transitions as a form of action, GCQ can also be used to predict observation changes driven by abstract-level variations. These predictions likewise exhibit long-range stability (Fig. 5, rows 3–4).

**Goal-directed planning.** Given a goal and an initial position, the GCQ can utilize the distance in the cognitive map to generate the most desirable action for the current step. After executing the action, a new observation is obtained, and this process is iterated, continuously reducing the distance to the goal in the cognitive map until the goal is reached (Fig. 3B, Fig. 6 rows, 1–2). The computation of the action at each step is of constant complexity.

**Inverse model.** Given a sequence of observations, the GCQ can first map them onto the cognitive map and then use goal-directed planning to determine the action or sequence of actions between adjacent observations, thus implementing the inverse model. The corresponding action sequence can be applied to the latent representation of another observation, and using the prediction capability, the generated sequence under this set of actions can be obtained (Fig. 6 rows, 3–4).

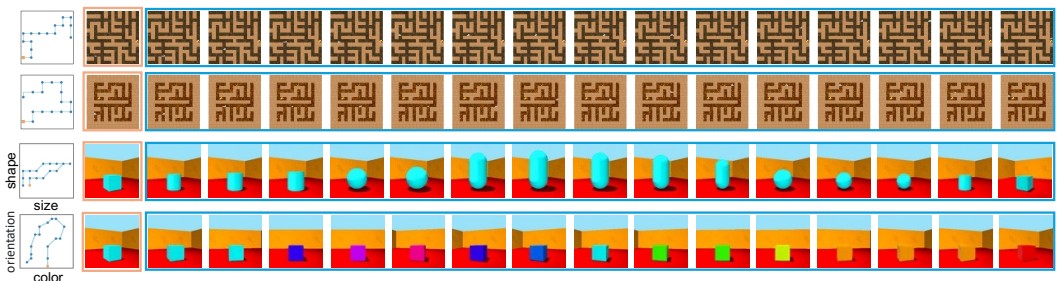

Figure 5: With the same setup as Fig. 4E, Rows 1–2: Prediction on the 2DMaze dataset. Rows 3–4: Prediction on the 3DShapes dataset.

Goal-directed planning

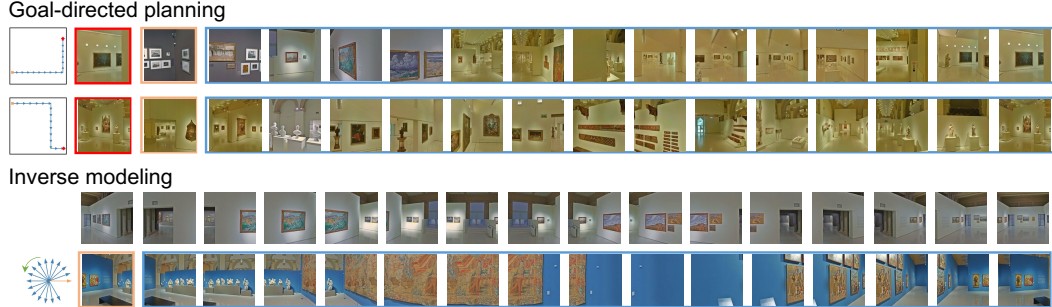

Figure 6: Rows 1–2: Goal-directed planning. The first patch shows the trajectory after planning, with orange indicating the starting point, red indicating the endpoint, and blue representing the planned route. The subsequent red-framed patches represent the endpoint, orange-framed patches represent the starting point, and blue-framed patches represent intermediate observations encountered during the process. Rows 3–4: Inverse modeling. The top row displays the given observation sequence. The bottom row starts with the first patch showing the action trajectory inferred from the observation sequence, with orange indicating the given initial observation, followed by the sequence generated based on the action trajectory.

# 6 Discussion

In this work, we introduced GCQ, a brain-inspired framework for compressing observation–action sequences into discrete, structured representations. GCQ uses continuous attractor dynamics to generate grid-like codewords, and selects them in an action-conditioned manner to capture both spatial and temporal dependencies. This spatiotemporal quantization process produces compact latent representations that serve as cognitive maps, enabling long-horizon prediction, goal-directed planning, and inverse modeling. Our experiments demonstrate that GCQ supports generalization across tasks while offering interpretability through its structured latent space.

**Insights for Neuroscience.** Beyond its practical performance, GCQ also offers a new computational hypothesis for the emergence of GCs in the brain. Traditionally, the formation of neurons with structured tuning properties was approached through handcrafted models [52], which provided only limited explanatory power. In contrast, the machine learning paradigm offers a data-driven framework: artificial neural networks are optimized to perform cognitive tasks, and their internal representations are analyzed to reveal emergent coding principles. Following this approach, prior studies have shown that GCs can arise when networks are trained to perform path integration under biologically constraints [53, 54, 55]. Subsequent work has emphasized the importance of predictive rather than reconstructive objectives [56] and extended the analysis to more general frameworks such as world models and predictive learning [57]. Efforts to induce disentangled representations through architectural or loss function constraints have further refined these insights [58, 59]. However, the robustness of grid-like pattern emergence remains debated [60], with some studies [61] suggesting that specific architectural features (e.g., one-hot inputs) are necessary.

Recent developmental findings add a new dimension to this discussion. Experiments show that toroidal activity patterns emerge in the medial entorhinal cortex even before sensory experience [62]. Intriguingly, such toroidal structures can be naturally modeled by bump attractors in CANNs. This suggests that the brain may possess preconfigured low-dimensional structures capable of bump activity, even prior to learning. Consistent with this, recent work has argued that GCs likely emerge from internal CANN mechanisms rather than from purely feedforward architectures [63].

This leads us to a novel hypothesis inspired by GCQ: GCs may arise not from optimizing networks, but from learning to map sensory experience onto a set of preexisting bump-based activity patterns. In GCQ, the codebook is defined by CANN-generated bumps before learning begins. The learning process then consists of associating observation–action sequences with combinations of these fixed codewords. Similarly, we speculate that the brain may use a fixed set of toroidal patterns—produced by CANNs—as a biological codebook. Through experience, the brain learns to map external sensory inputs onto these internal structures, endowing them with meaning and interpretability, allowing for the decoding of grid-like patterns. This perspective suggests a unified model of how the brain may simultaneously achieve compression and semantic organization of sensory information.

**Static Setting.** We also evaluated GCQ in a static setting, where the sequence length is 1. In this case, we compared GCQ with VQ-VAE on ImageNet [64], finding that GCQ suffered minimal performance degradation. This suggests that the use of a fixed codebook does not significantly harm performance on static tasks. For further details, refer to Appendix A.

**Scalability of Action.** Our current work utilizes 2D attractors, where each has 5 potential transitions (four shifts and one stationary). With such $m$ CANNs, the model can represent $5^m$ distinct actions. If we use $P$-dimensional attractors, the number of states per CANN will become $2P+1$, yielding a total action space of $(2P+1)^5$. Therefore, GCQ can be scaled to higher-dimensional action spaces by adjusting both $m$ and $P$. For the experiments in this paper, which involve relatively low-dimensional actions, 2D attractors are sufficient.

**Future Work.** A promising direction for advancing GCQ lies in enabling the encoder and decoder to process entire sequences holistically, rather than treating each sequence element independently. Incorporating ViTs with spatial-temporal attention could serve as an effective approach toward this goal. Moreover, scaling GCQ to larger and more diverse datasets would facilitate a deeper investigation into its generalization capabilities and robustness across a broader range of tasks and domains.

## Acknowledgments

This work was supported by the National Natural Science Foundation of China (no. T2421004 to S.W.), the Science and Technology Innovation 2030-Brain Science and Brain-inspired Intelligence Project (no. 2021ZD0200204, S.W.).

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

# A  Static Setting

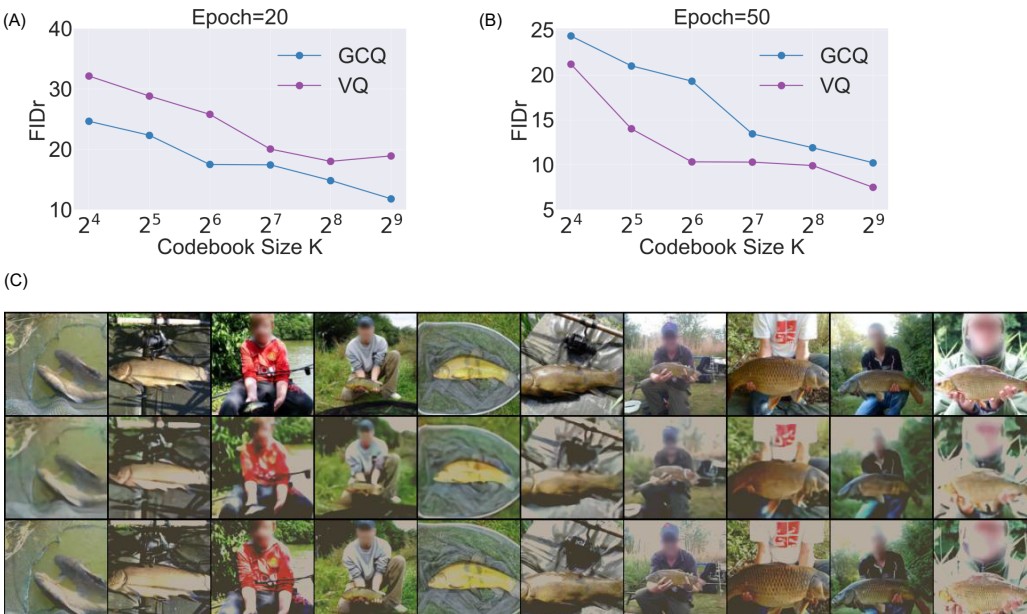

(A)

(B)

(C)

Figure 7: To evaluate the performance of GCQ in a static setting, we conducted experiments on the CIFAR-10 and ImageNet datasets. Figures (A) and (B) show that, for both GCQ and VQ, the FIDr decreases as the codebook size K increases. At 20 training epochs, GCQ outperforms VQ, whereas at 50 epochs VQ achieves better results. Figure (C) compares the reconstructions produced by GCQ and VQ on ImageNet: the first row shows the original images, the second row the reconstructions obtained with GCQ, and the third row those obtained with VQ.

# B  Experiment Details

Here are the hyperparameters used in the experiment. All programs run on an NVIDIA A100-SXM4-80GB. The experiments reported in this paper, including the ViT, ResNet, and Hybrid networks, required 8-12 hours of training each. For ViT and hybrid architectures, we trained for 40 epochs with a learning rate of 1e-4; for the ResNet network, we trained for 100 epochs with a learning rate of 3e-4. All training runs used the Adam optimizer.

# C  Python Implementation

Our Python implementation of GCQ is fully vectorized, relying exclusively on matrix operations without any for loops. This design makes it highly amenable to parallelization.

```python
def forward(self, latents: Tensor, label: Tensor) -> Tuple[Tensor,
    Tensor]:
    """
    Vector quantization for sequence data with parallel processing
    support for batch dimensions

    Args:
        latents: Tensor of shape [B x S x D x H x W]
        label: Tensor of shape [B x S x 4] representing the change
    from current frame to next frame

    Returns:
        quantized: Quantized tensor of shape [B x S x D x H x W]
        vq_loss: Vector quantization loss
```

```python
        """
        B, S, D, H, W = latents.shape
        N = H * W // 4
        device = latents.device

        # Reshape to processable form
        latents = latents.view(B, S, D, 4, N).permute(0, 3, 4, 1, 2).
        contiguous()  # [B, 4, N, S, D]
        factor_latents = latents.view(B, 4, N, S * D)

        # Construct expanded embeddings without loops
        # label: [B, S, 4] -> rearrange to [B, 4, S]
        shift_amounts = (label.permute(0, 2, 1).long() * self.step)  # [B,
         4, S]
        # Construct indices for rolling: for each shift_amount against
        self.embedding_weight
        # First generate indices [K], then calculate new indices via
        broadcasting: new_idx = (arange(K) - shift) % K
        k_idx = torch.arange(self.K, device=device).view(1, 1, 1, self.K)
         # [1,1,1,K]
        # After expanding shift_amounts: [B,4,S,1]
        rolled_indices = (k_idx - shift_amounts.unsqueeze(-1)) % self.K  #
         [B, 4, S, K]
        # Using rolled_indices to extract new embeddings from
        embedding_weight, resulting shape [B,4,S,K,D]
        expanded_embed = self.embedding_weight[rolled_indices]  # [B,4,S,K
        ,D]
        # Adjust dimensions: exchange [K] and [S] positions before merging
         S and D
        expanded_embed = expanded_embed.permute(0, 1, 3, 2, 4).contiguous
        ()  # [B,4,K,S,D]
        # Finally reshape to [B, 4, K, S*D]
        expanded_embedding = expanded_embed.view(B, 4, self.K, S * D)

        # Calculate nearest neighbor indices
        # factor_latents: [B,4,N,S*D]; expanded_embedding: [B,4,K,S*D]
        A = factor_latents  # [B,4,N,S*D]
        B_expand = expanded_embedding  # [B,4,K,S*D]
        A_sq = (A ** 2).sum(dim=-1, keepdim=True)  # [B,4,N,1]
        B_sq = (B_expand ** 2).sum(dim=-1).unsqueeze(-2)  # [B,4,1,K]
        cross = 2 * torch.matmul(A, B_expand.transpose(-1, -2))  # [B,4,N,
        K]
        dist = A_sq + B_sq - cross  # [B,4,N,K]
        encoding_inds = dist.argmin(dim=-1)  # [B,4,N]

        # Sample vectors from expanded_embedding according to indices
        using torch.gather
        # expanded_embedding shape [B,4,K,S*D], sampling on dim=2
        encoding_inds_exp = encoding_inds.unsqueeze(-1).expand(-1, -1, -1,
         S * D)  # [B,4,N,S*D]
        embedding_results = torch.gather(expanded_embedding, 2,
        encoding_inds_exp)  # [B,4,N,S*D]

        commitment_loss = F.mse_loss(embedding_results.detach(),
        factor_latents)
        embedding_results = factor_latents + (embedding_results -
        factor_latents).detach()  # [B,4,N,S*D]
        embedding_results = embedding_results.view(B, 4, N, S, D)
        embedding_results = embedding_results.permute(0, 3, 4, 1, 2).
        contiguous().view(B, S, D, H, W)
        return embedding_results, commitment_loss
```

