# OpenReview forum: "Vector Quantization in the Brain: Grid-like Codes in World Models"
_NeurIPS.cc/2025/Conference — NeurIPS 2025 spotlight_

### Official Review · Reviewer_NYvu · 2025-06-26

**Clarity:** 3
**Significance:** 3
**Originality:** 4
**Rating:** 5
**Confidence:** 4

**Summary:**

The authors propose grid-like code quantization (GCQ) as a brain-inspired method for compression of observation-action sequences into discrete tokens through grid-cell-like bumps in continuous attractor neural networks. GCQ builds an action-conditioned codebook for (spatio-)temporal data rather than a fixed codebook on static inputs. They provide experiments on 2DMaze, Google Street View, MPI3D, 3DShapes and static CIFAR/ImageNet tasks to show how GCQ outperforms standard VQ baselines (which are coupled with a U-Net for temporal sequences). GCQ also demonstrates superior long-horizon prediction, goal-directed planning and inverse modeling compared to the baselines. The authors connect their model to neuroscience literature and provide a theoretical perspective on grid-cell emergence in the brain.

**Questions:**

- Can you confirm that the number of CANNs required $m$ scales logarithmically with the number of possible actions $|A|$ that can be represented by the GCQ model?
- What is the computational cost of the matching, is it $O(mKn)$ (for $n$ sequence length, $m$ CANNs and $K$ attractors)? In that case, matching cost would become prohibitive for video-scale resolutions and $n>>10$. Is there a way around this?

**Ethical Concerns:**

["NO or VERY MINOR ethics concerns only"]

**Final Justification:**

I appreciate the additional experiments and explanations that the authors provided during the rebuttal. I would encourage the authors to include their explanation for scaling the dimensionality of attractors $P$ and the number of CANNs $m$ in the paper, I found it very instructive. I am happy to keep my score of 5 in support of this work.

**Limitations:**

- The paper does not address how to handle continuous action spaces, or scalability to very large discrete action spaces.
- GCQ depends critically on action labels, which limits its applicability on real-world data that doesn't have action labels.

**Quality:**

3

**Strengths And Weaknesses:**

**Strengths**
- GCQ integrates spatial and temporal quantization into a unified model for spatiotemporal compression via an action-conditioned codebook (avoiding two-stage pipelines like VQ+U-Net).
- Actions correspond to explicit bump shifts in the attractor network which yields a cognitive map with interpretable latent structure that is conducive to inverse modeling.
- Empirical results show strong performance of the method on relevant benchmarks, while also still performing well on static image benchmarks. Particularly, the long-horizon predcition and planning is impressive.
- The authors offer a data-driven hypothesis (that is grounded in strong performance on real ML benchmarks) for how grid-cell codes exist in attractor networks and support learning sensory representations that are effective for predictions and planning.
- It appears that GCP is compatible with any encoder-decoder backbone and the authors present an ablation of different backbones that shows that the ViT backbone strives a good trade-off between efficiency and performance.

**Weaknesses**:
- GCQ requires explicit, discrete action labels and an injective mapping $|A| \leq 5^m$ which limits applicability where actions are unknown or continuous.
- It is not clear how fine-grained actions or high-dimensional control can be captured with the fixed bump shifts without scaling $m$. This scaling of $m$ (and thus, compute) on more  complex tasks  is not studied.
- Evaluated primarily on small-scale image sequences and synthetic mazes; scalability to complex, real-time video or continuous-control domains remains untested (e.g., BAIR robot pushing, Kinetics-600, driving benchmarks).
- Impact of noise or action-mislabeling on the template-matching quantizer is not analyzed.

Minor:
- Line 276: "GCQ converged faster" is not quite right because the performance of GCQ has not converged yet at the point of comparison. It simply performed better at lower training time but was eventually surpassed by VQ.

---

> ### Author Rebuttal · Authors · 2025-07-29
>
> We sincerely thank the reviewer for your positive and insightful feedback. We are very encouraged by your recognition of our work's key strengths, including our unified GCQ model that integrates spatial and temporal quantization via an action-conditioned codebook, the interpretable latent structure yielding a cognitive map conducive to inverse modeling, the strong empirical performance, particularly on long-horizon prediction and planning, and our well-supported, data-driven hypothesis for the function of grid-cell codes. Your accurate summary and positive assessment are greatly appreciated.
>
> ---
>
> ### **Response to Weaknesses**
>
> 1.  The reviewer correctly points out that the version of GCQ presented in this manuscript is designed for tasks with explicit, discrete action labels. We totally agree that extending GCQ to continuous or unknown action spaces is a valuable research direction. In fact, we have already begun theoretical and experimental explorations of GCQ variants for these scenarios.
>     *   For continuous action spaces, we plan to use a variant to replace the discrete bump shifts with rotation matching based on continuous rotation angles  (similar to RoPE method) .
>     *   For unlabeled action spaces, we leverage the strong temporal correlation between adjacent frames in a sequence as a supervisory signal, enforcing that the codes for proximate frames should also be closely related.
>
>     Our preliminary results indicate that these variants are not only feasible but also achieve promising performance. We consider this a significant avenue for future work and plan to report our findings in a subsequent publication.
>
> 2.  The reviewer raises an important point about scaling to finer-grained or higher-dimensional action spaces. There are two primary mechanisms to scale the representative capacity of GCQ:
>     *   Increasing the number of CANNs ($m$).
>     *   Increasing the dimensionality of the attractors ($P$).
>     As described in Line 123, our current work utilizes 2D attractors, where each has 5 potential transitions (four shifts and one stationary). With $m$ such CANNs, the model can represent $5^m$ distinct actions. If we use P-dimensional attractors, the number of states per CANN will become $(2P+1)$, yielding a total action space of $(2P+1)^m$. Therefore, GCQ can be scaled to higher-dimensional action spaces by adjusting both $m$ and $P$. For the experiments in this paper, which involve relatively low-dimensional actions, 2D attractors are sufficient.
>
> 3.  We concur with the reviewer that evaluating GCQ on complex domains such as real-time video and continuous control is a great direction for extending our work. As mentioned in our response to point 1, the variant using continuous rotation matching is well-suited for continuous control tasks and it is designed with an efficient matching algorithm to maintain computational performance. We have already conducted preliminary experiments on datasets like TartanDrive, RECON, SCAND, and HuRoN with positive results, sorry we are unable to present figures here due to the rebuttal policy. The datasets suggested by the reviewer (e.g., BAIR robot pushing, Kinetics-600) are highly relevant, and we will consider them for future investigation.
>
> 4.  We thank the reviewer for this excellent suggestion to test the model's robustness. We have performed supplementary experiments to analyze the impact of noisy action labels. The table below shows the FIDp score as a function of the noise level,
>
>     |    Noise level  |  1%  |  2%   | 4% | 8% |16%|
>     | ----------- | ------ | ------ | ------ |------ |------ |
>     | GCQ (112M) | 44 | 44 | 46| 52| 58 |
>
>     We believe these results effectively demonstrate the robustness of our method. We commit to including this analysis and discussion in the final version of the paper.
>
> 5.  *Minor (Line 276):* We agree with the reviewer's observation and appreciate the suggestion for more precise phrasing. The statement "GCQ converged faster" is indeed not entirely accurate at that point of comparison. In the final version, we will revise this to state that "GCQ demonstrated superior performance in the early stages of training" and will consider adding a plot of the training loss over time to provide clearer evidence of its convergence behavior.
>
> ---
>
> ### **Response to Questions**
>
> 1.  Yes, we can confirm that the number of CANNs ($m$) scales logarithmically with the number of representable actions.  To provide a more detailed clarification: each 2D CANN is built upon two orthogonal bases, and each base supports transitions in two opposing directions. Together with the option to remain stationary, this yields a total of 5 discrete transitions per CANN (i.e., 2 bases × 2 directions + 1 stationary option). Since the $m$ CANNs operate independently, the total set of possible actions is the Cartesian product of the individual action sets. This results in a total of $5^m$ unique representable actions. Consequently, the number of CANNs required, $m$, is related to the total number of actions, $|A|$, by $m = \log_5(|A|)$.
>
> 2.  The reviewer's analysis of the computational cost as $O(mKn)$ is correct. However, we would like to clarify that this matching operation is highly parallelizable. When executed on a GPU, the computations across $m$,  $K$, and $n$ can be performed in parallel, which significantly mitigates the computational burden and prevents it from becoming a bottleneck, even for larger-scale inputs. We include the relevant codes for the matching algorithm here:
>
> ```python
> def forward(self, latents: Tensor, label: Tensor) -> Tuple[Tensor, Tensor]:
>     """
>     Vector quantization for sequence data with parallel processing support for batch dimensions
>
>     Args:
>         latents: Tensor of shape [B x S x D x H x W]
>         label: Tensor of shape [B x S x 4] representing the change from current frame to next frame
>
>     Returns:
>         quantized: Quantized tensor of shape [B x S x D x H x W]
>         vq_loss: Vector quantization loss
>     """
>     B, S, D, H, W = latents.shape
>     N = H * W // 4
>     device = latents.device
>
>     # Reshape to processable form
>     latents = latents.view(B, S, D, 4, N).permute(0, 3, 4, 1, 2).contiguous()  # [B, 4, N, S, D]
>     factor_latents = latents.view(B, 4, N, S * D)
>
>     # Construct expanded embeddings without loops
>     # label: [B, S, 4] -> rearrange to [B, 4, S]
>     shift_amounts = (label.permute(0, 2, 1).long() * self.step)  # [B, 4, S]
>     # Construct indices for rolling: for each shift_amount against self.embedding_weight
>     # First generate indices [K], then calculate new indices via broadcasting: new_idx = (arange(K) - shift) % K
>     k_idx = torch.arange(self.K, device=device).view(1, 1, 1, self.K)  # [1,1,1,K]
>     # After expanding shift_amounts: [B,4,S,1]
>     rolled_indices = (k_idx - shift_amounts.unsqueeze(-1)) % self.K  # [B, 4, S, K]
>     # Using rolled_indices to extract new embeddings from embedding_weight, resulting shape [B,4,S,K,D]
>     expanded_embed = self.embedding_weight[rolled_indices]  # [B,4,S,K,D]
>     # Adjust dimensions: exchange [K] and [S] positions before merging S and D
>     expanded_embed = expanded_embed.permute(0, 1, 3, 2, 4).contiguous()  # [B,4,K,S,D]
>     # Finally reshape to [B, 4, K, S*D]
>     expanded_embedding = expanded_embed.view(B, 4, self.K, S * D)
>
>     # Calculate nearest neighbor indices
>     # factor_latents: [B,4,N,S*D]; expanded_embedding: [B,4,K,S*D]
>     A = factor_latents  # [B,4,N,S*D]
>     B_expand = expanded_embedding  # [B,4,K,S*D]
>     A_sq = (A ** 2).sum(dim=-1, keepdim=True)  # [B,4,N,1]
>     B_sq = (B_expand ** 2).sum(dim=-1).unsqueeze(-2)  # [B,4,1,K]
>     cross = 2 * torch.matmul(A, B_expand.transpose(-1, -2))  # [B,4,N,K]
>     dist = A_sq + B_sq - cross  # [B,4,N,K]
>     encoding_inds = dist.argmin(dim=-1)  # [B,4,N]
>
>     # Sample vectors from expanded_embedding according to indices using torch.gather
>     # expanded_embedding shape [B,4,K,S*D], sampling on dim=2
>     encoding_inds_exp = encoding_inds.unsqueeze(-1).expand(-1, -1, -1, S * D)  # [B,4,N,S*D]
>     embedding_results = torch.gather(expanded_embedding, 2, encoding_inds_exp)  # [B,4,N,S*D]
>
>     commitment_loss = F.mse_loss(embedding_results.detach(), factor_latents)
>     embedding_results = factor_latents + (embedding_results - factor_latents).detach()  # [B,4,N,S*D]
>     embedding_results = embedding_results.view(B, 4, N, S, D)
>     embedding_results = embedding_results.permute(0, 3, 4, 1, 2).contiguous().view(B, S, D, H, W)
>     return embedding_results, commitment_loss
> ```
>   As illustrated in the code, we have replaced all iterative loops with matrix operations so the time complexity does not become a bottleneck in practice. However, we acknowledge that the space complexity does scale with the input size. Therefore, parameters such as $m$ and $K$ should be adjusted based on the available hardware resources during deployment to strike a balance between performance and memory footprint.
>
> ---
>
> ### **Response to Limitations**
>
> *   As detailed in our **Response to Weaknesses 1**, we are actively developing GCQ variants to address continuous action spaces and scenarios where action labels are unavailable. While these extensions are beyond the scope of the current paper, they are a primary focus of our future research.
> *   The strategy for scaling to very large discrete action spaces by adjusting the number of CANNs ($m$) and the attractor dimensionality ($P$) is outlined in our **Response to Weaknesses 2**.
> ---
>
> Once again, we thank the reviewer for your positive assessment and constructive suggestions. We are particularly grateful for the suggestions regarding GCQ variants for continuous and unlabeled actions, which align perfectly with the focus of our future research. We hope our responses have addressed all your concerns. We welcome any further questions or suggestions.

---

> > ### Comment · Reviewer_NYvu · 2025-08-05
> >
> > I thank the authors for their extensive replies to all reviews. I appreciate the additional experiments on robustness to noise and their explanations regarding complexity and parallelization. I would encourage the authors to include their explanation for scaling the dimensionality of attractors $P$ and the number of CANNs $m$ in the paper, I find it very instructive. I am happy to keep my score of 5 in support of this work.

---

> > > ### Author Response · Authors · 2025-08-05
> > >
> > > Thank you for your encouraging feedback on our rebuttal. We sincerely appreciate the time and effort you have dedicated to reviewing our work.
> > >
> > > We are pleased that you found our additional experiments on robustness to noise and our explanations regarding complexity, parallelization and scaling to be valuable. We commit to incorporating the supplementary experiments and detailed explanations into the final version.
> > >
> > > Thank you once again for your positive assessment and for maintaining your score in support of our work.

---

> > > > ### Comment · Reviewer_NYvu · 2025-08-05
> > > >
> > > > Thank you for your quick response, all the best for your exciting research!

---

### Official Review · Reviewer_wnzM · 2025-06-28

**Clarity:** 3
**Significance:** 3
**Originality:** 3
**Rating:** 5
**Confidence:** 3

**Summary:**

The paper introduces Grid-like Code Quantization (GCQ) which observation–action sequences into discrete codes by leveraging grid-pattern attractor dynamics. GCQ compresses both spatial and temporal information via an action-conditioned codebook: codewords are generated from continuous attractor neural networks and are selected dynamically based on the executed action. Experimental results show that GCQ delivers significant gains in long-horizon prediction, goal-directed planning, and inverse modeling tasks.

**Questions:**

Please see strengths and weaknesses.

**Ethical Concerns:**

["NO or VERY MINOR ethics concerns only"]

**Final Justification:**

I have carefully reviewed the rebuttal discussion and additional experiments, which address my concerns, specifically:
- The author has clarified the design of the VQ model, which was unclear in the initial paper, and has promised to update this in the revised paper.
- The author has committed to adding a description of the typical VQ configuration before introducing GCQ, ensuring such a comparison will benefit readers.
- The author has conducted additional experiments (planning success rate), which I would have liked to see in the rebuttal.

I also reviewed the author's rebuttal to the other reviewer. Based on these responses, I raise the overall score to 5 and adjust the quality and clarity scores accordingly.

**Limitations:**

Yes

**Quality:**

3

**Strengths And Weaknesses:**

### **Strengths**

The proposed Grid-like Code Quantization (GCQ) for compressing observation–action sequences into discrete representations is novel. As discussed in the experiments, this quantization delivers clear advantages for long-horizon prediction and goal-directed planning through an inverse-model interpretation.

### **Weaknesses**

The method and experimental sections are overly generic and omit several important implementation details.

**Method**

- Section 3 states that, given an input $I$, GCQ maps $I$ onto one of $N^2$ attractors that make up the codebook. Equation (6) then defines two orthogonal action bases whose indices align with those attractors, yielding an action-conditioned codebook. The authors claim that GCQ compresses both spatial and temporal information.

  For comparison, consider VQ. Given an embedding $z$, VQ usually partitions z into m subvectors, $z = \[z_1, …, z_m\]$, each of which is mapped to its closest codeword (product quantization). When a sequence of observations is quantized, consecutive observations may share codewords and thus capture temporal correlation. Because the manuscript does not specify the exact VQ variant, it is unclear how the baselines were configured. Could the authors specify the set-up used for VQ?

- Figure 2 (“Schematic of GCQ: latent state composed of m = 3 codes”) is difficult to follow. Equation (6) implies that an action corresponds to a transition between attractors (or observations). If a single observation is represented by m attractors, how is the composite action defined? Please clarify how multiple codes are jointly updated when one atomic action is applied.

To aid readability, it would help to first describe a typical VQ configuration and then highlight how GCQ deviates from it. This parallel presentation would create a more intuitive bridge for the machine-learning community.

**Experiments**: The experimental setup lacks critical details for adjust the advantage and avaluate the comparision.

Table 1 is the sole quantitative result, yet it does not convincingly demonstrate the efficacy of GCQ. Additional metrics—for example, reconstruction error and planning success rate—would strengthen the empirical evidence.


 - Please clarify what “model size” denotes in Figure 4a. In VQ-VAE the codebook size is crucial, so does your reported size include the codebook parameters or only the encoder–decoder backbone? How do you ensure the comparison is fair?

- Line 204-205: Because the action embedding builds on Latent Action Policies (LAPO), it is unclear that **goal-directed planning and Inverse model properties** are gained stem from GCQ and which stem from LAPO. A direct LAPO baseline is therefore necessary.

- Goal-directed planning: Line 229 states that a goal and an initial position are provided. Is goal conditioning seen during training, or is it only applied at test time?

**0verall:**
- I think the method is innovative and potentially very impactful, but the manuscript needs substantial improvements in clarity and completeness.

- I apologize for any misunderstandings; I am happy to revisit my evaluation and adjust the score once these points are clarified.

---

> ### Author Rebuttal · Authors · 2025-07-26
>
> Thank you for the valuable and constructive comments, the recognition of the novelty of our work on using GCQ for compressing observation–action sequences,  and the acknowledgement of our method’s potential advantages for long-horizon prediction and goal-directed planning.
>
>    ---
>
> ### **Response to Weaknesses**
>
> #### **Method**
>
> * We would like to clarify the set-up used for the VQ baseline. As stated in the Baselines section (line 201) of the manuscript, the model referred to as “VQ” is in fact a two-stage architecture, consisting of a VQ-VAE [1] followed by a UNet. We adopted this term as a shorthand for brevity, and we apologize if this naming led to any confusion.
>
>   &nbsp;
>
>   The rationale for this design is as follows: This two-stage architecture is essential for our world modeling task, which requires processing sequences of observations and actions. The VQ-VAE component handles the compression of static images into discrete latent codes ($o_t\to z_t$). However, to model the evolution of these codes over time, an explicit temporal model is necessary. The UNet's role, therefore, is to explicitly model the temporal dynamics across the sequence of these latent codes ($(z_t,a_t)\to z_{t+1})$. It is worth noting that unlike the VQ baseline model, our GCQ does **not** rely on a two-stage architecture, as it integrates spatiotemporal compression within a unified framework— which represents a key contribution of our work.
>
>   &nbsp;
>
>    It is important to note that this two-stage design is a standard and widely adopted practice in recent world modeling literature. Many state-of-the-art methods first use a VQ-VAE or VAE for spatial compression and then employ a separate, powerful sequence model (like a Transformer, RNN or UNet) to capture temporal dependencies [2][3]. Our choice of the baseline model is therefore consistent with established approaches in the field.
>
>   &nbsp;
>
>    To ensure this is clear to all readers, we will revise the manuscript to replace the "VQ" shorthand with the full term "**VQ+UNet**" throughout the text, tables, and figures. We hope this explanation clarifies our experimental setup, and we thank you for helping us improve the precision of our paper.
>
>   &nbsp;
>
>    In our main experiments, the specific UNet architecture we used is identical to the one proposed in the LAPO paper . To further strengthen our comparison and demonstrate the robustness of our approach, we also conducted additional experiments with a **VQ+transformer** baseline. For this, we replaced the UNet with a Transformer whose architecture is based on TransDreamer [2]. The results, summarized below, show that our proposed method outperforms both the UNet-based and the Transformer-based baselines. The table below shows the FIDp score as a function of the prediction step, similar to the analysis in Fig. 4D. Lower is better.
>
>   |    Prediction Steps  |  1  | 2   | 4 | 8 |16|32|
>   | ----------- | ------ | ------ | ------ |------ |------ |------ |
>   | VQ+Unet (96M) | 51| 78| 115| 149| 159| 164|
>   | VQ+Transformer (121M) | 61 | 64 | 71 | 79 | 88 | 111 |
>   | GCQ (112M) | 42| 44| 43| 44|44 |44|
>
> * Let us first clarify the notation. In the paper, each frame is compressed to $m$ codes and $e_j$ denotes the $j$-th code ($j=1,...,m$). $a_i$ refers to the action vector applied to all codes at time step $i$. When an action $a_i$ is applied to a latent state composed of $m$ codes, each code $e_j$ undergoes a transition determined by the $j$-th component of $a_i$, denoted $a_{i,j}$. To make this process clearer, we have revised the manuscript and replaced Equation (7) with the following:
>    $$
>    e_j \oplus a_{1:n-1} = \lbrace e_j, e_j + a_{1,j}, \dots, e_j + a_{1:n-1,j} \rbrace \tag{7*}
>    $$
>
>   &nbsp;
>
>    As an example, for a latent state composed of $m = 3$ codes, consider a moving agent. We would first manually define the overall bump transitions corresponding to different movements. For instance, "move left" is defined as $a_{left} = [a_\theta^+, 0, 0]$ (the first bump moves slightly in the positive theta direction, while the other two bumps remain stationary). At time step $i$, the latent code is $s_i$, and the movement at the $i$-th time step is to the left, i.e., $a_i = a_{left}$. Then, $s_{i+1} = s_i + a_i$ (the first bump of $s_i$ moves slightly in the positive theta direction, while the other two bumps remain stationary).We hope this clarification and the updated equation address your concern.
>
>  This is a great suggestion—we will add a description of the typical VQ configuration prior to introducing GCQ, as we agree that such a comparison would benefit readers.
>
> ---
>
> #### **Experiments**
>
> First, we would like to point out that Table 1 reports PSNR—a widely used metric for **reconstruction error** that serves as a perceptual measure of image compression performance. Also the FID scores reported in Table 1, Figure 4(A)(B)(C)(D), and Figure 7(A)(B) are also standard and widely adopted metrics for assessing the quality of generated images.
>
> Additionally, we have conducted supplementary experiments measuring the planning success rate, and the results are shown in the table below(2DMaze dataset):
>
>
> |             | 542M   | 307M   | 140M   |
> | ----------- | ------ | ------ | ------ |
> | seen goal   | 97/100 | 93/100 | 68/100 |
> | unseen goal | 91/100 | 88/100 | 65/100 |
>
>
> We will incorporate these results into the final version of the paper. We hope these additional experiments help substantiate the effectiveness of GCQ.
>
> 1. In the case of GCQ, the model size refers to the encoder–decoder backbone, as the codebook in GCQ is non-trainable and does not contain learnable parameters. In contrast, the VQ baseline requires both a VQ-VAE and an additional network (e.g., UNet) to complete the experimental tasks—VQ-VAE alone is insufficient. Therefore, we reported the total parameter count of this two-stage pipeline. Importantly, GCQ and VQ are matched in terms of compression ratio and codebook size, ensuring a fair comparison. The reported model sizes reflect all learnable parameters in the entire systems.
>
> 2. In fact, **GCQ alone** is fully capable of learning action embeddings and accomplishing all tasks in our experiments, including long-horizon prediction, goal-directed planning, and inverse modeling. This is a key advantage of GCQ over VQ. While VQ requires additional modules—such as the UNet from LAPO—to construct a two-stage world model, GCQ integrates spatiotemporal compression into a single, unified framework without relying on any auxiliary architecture.
>
>    Importantly, LAPO components are used **only** for the VQ-based baselines and play no role whatsoever in the implementation, training, or evaluation of GCQ. We hope this clarification resolves any potential misunderstanding.
>
> 3. For the GSV dataset, we used images seen during training as the goals for goal-directed planning. In contrast, for the 2DMaze dataset, we used goals that were unseen during training. This approach was chosen because the GSV dataset has a complex data distribution, and the limited amount of data used for our training makes generalization challenging. In our future work, we plan to leverage a larger dataset to enable planning to unseen locations within the GSV dataset.
>
> ---
>
> Finally, we once again thank you for recognizing the innovative nature and potential impact of our work. We hope our responses address your concerns and misunderstandings regarding both the methodology and experimental design. We also appreciate your constructive suggestions, and we commit to incorporating the corrected equation and additional experiments into the final version of the paper.
>
> ---
>
> [1] Neural discrete representation learning.
>
> [2] Transdreamer: Reinforcement learning with transformer world models.
>
> [3] Efficient World Models with Context-Aware Tokenization.

---

> > ### Comment · Reviewer_wnzM · 2025-08-03
> > **Thank you for the clarification and additional experiment.**
> >
> > Thank you to the author for the detailed response to my concern. I have carefully reviewed the rebuttal discussion and additional experiments, which address my concerns. I also reviewed the author's rebuttal to the other reviewer. Based on this, I will raise the overall score to 5 and adjust the quality and clarity scores accordingly in the next phase of the review process.
> >
> > Regarding the point, "we will add a description of the typical VQ configuration prior to introducing GCQ," it would be even better if the author could also provide the pseudocode for the VQ-based method and GCQ used in the experiments.
> >
> > I look forward to reading the revised version and the extension of GCQ to continuous or unknown action spaces, as discussed in response to Reviewer NYvu.
> >
> > Wishing the author all the best with this submission.

---

> ### Author Response · Authors · 2025-08-03
> **Grateful for your positive feedback and confirmation that concerns are addressed**
>
> We would like to express our sincere gratitude for your thorough review of our rebuttal and additional experiments. We are glad to hear that our response has addressed your concerns, and we deeply appreciate you raising the overall score for our paper.
>
> Regarding your valuable suggestion, we confirm that we will add the pseudocode for both the baseline VQ-based method and our GCQ in the revised version of the paper. We agree this will significantly enhance the clarity of our work.
>
> We are also very encouraged by your interest in our future work on extending GCQ to continuous or unknown action spaces. Your feedback gives us more confidence in pursuing this direction.
>
> Once again, thank you for your support and constructive guidance throughout this process. We look forward to preparing the revised version.

---

### Official Review · Reviewer_imb7 · 2025-07-09

**Clarity:** 3
**Significance:** 4
**Originality:** 4
**Rating:** 5
**Confidence:** 3

**Summary:**

Grid-like Code Quantization or GCQ is introduced as a brain-inspired alternative to conventional vector-quantization schemes. Instead of learning a static codebook, GCQ derives its discrete codes from bumps in continuous-attractor neural networks or CANNs, whose periodic lattice naturally mirrors the grid-cell patterns seen across the cortex. By letting actions shift these bumps, the method performs action-conditioned quantization of entire observation–action sequences, jointly compressing space and time into a single latent representation that functions as a world model and a cognitive map.

GCQ retains the familiar encoder, quantizer, and decoder pipeline but replaces frame-wise template matching with sequence-to-sequence matching against an action-conditioned codebook. Each latent slot is tied to one CANN; valid actions translate the bump within that CANN, so planning becomes a search for a legal bump trajectory. The authors test three backbone families, finding a ViT encoder-decoder comes out on top. Across four vision datasets GCQ matches or surpasses a two-stage baseline that couples VQ-VAE with a UNet dynamics model, which isolates the strength of the CANN-based codebook. Because its latent map is intrinsically stable, GCQ’s prediction FID stays flat as rollout length grows, whereas the baseline blurs quickly. The same map also enables constant-time goal-directed planning and an inverse-model that infers the action sequence connecting two observations. Ablations show the ViT variant attains the best reconstruction (PSNR) and prediction quality with far fewer parameters than a ResNet counterpart.

Beyond engineering gains, the authors frame GCQ as a new hypothesis for how grid codes may emerge based on recent debates in the field between learned representations versus baked-in architectural constraints. Rather than being learned, the toroidal bump manifold could be pre-wired, with learning devoted to mapping sensory streams onto these pre-existing codewords. This view reconciles recent developmental data showing toroidal structure before experience, and it suggests that efficient compression and semantic organization may share a single neural substrate. Future work will scale GCQ to larger datasets and explore sequence-level ViT architectures, while probing the limits of fixed codebooks in more complex settings.

**Questions:**

I would welcome a clearer picture of how the proposed model relates to canonical hippocampal place-cell phenomena. Right now the paper analyses only grid-like bumps; showing receptive-field heat-maps or other evidence of place-like units—or explaining why such units do not emerge—would ground the biological claims and sharpen the contribution. Relatedly, the mapping from environment actions to CANN displacements is currently hard-coded. Please either supply a short empirical justification for this choice (e.g., ablation showing that a learned mapping offers no benefit) or re-frame it explicitly as future work so that readers understand the design trade-off. In Section 3, the three pages of continuous-attractor derivations interrupt the narrative flow; moving those equations into Methods or an appendix and replacing or adding to Figure 2 with a simpler schematic (perhaps with an inset zoom on the “multi-CANN codebook to decoder” path) would greatly improve readability. Finally, the sentence that cites “instability issues commonly seen in current world models” needs a reference, and the very last paragraph would read more professionally if phrased less casually.

Addressing the first two points (place-cell analysis and a justification (or learning) of the action-to-CANN mapping) would directly increase the perceived biological and methodological rigor, lifting my Quality and Significance scores by at least one notch. Conversely, if the final version retains the current ambiguity on those fronts, my overall rating may drop to the “borderline” range despite the paper’s conceptual appeal.

**Ethical Concerns:**

["NO or VERY MINOR ethics concerns only"]

**Final Justification:**

In light of the very useful modifications made during the rebuttal of my review for this paper, I would like to update my overall score from a 4 to a 5. In particular, a more rigorous defense of the use of hard-coded actions with an ablation study and further discussion, as well an appropriate rebuttal for my question regarding place cells, influenced my decision. I also felt that the discussion with reviewer wnzM regarding the VQ-VAE setup was particularly helpful, and that the VQ+Transformer baseline addition by the authors better situates this work in the current literature. Finally, though I cannot currently review the improvements that were made in response to my critiques on the formatting, references, and Figure 2, I appreciate that the authors took on these updates and I believe that the paper is stronger with these and the other changes.

**Limitations:**

There is no broader societal impact to consider in this work and the paper is correct to not address this point.

The paper could do a better job at considering limitations in the hard-coding of action sets as mentioned in previous sections of this review and could in general situate itself more productively between the two aims of world modelling and constructing a biologically-plausible computational cognitive map. If the aim is more to be a strong world modelling model, then the paper should have much more robust benchmarks against the existing cited world model papers, particularly on downstream RL tasks. If the aim is to be a more biologically-plausible grid cell model, then the paper should carefully consider and explain certain decisions such as offline learning, completely separated CANN modules, and hard-coded action sets.

**Paper Formatting Concerns:**

There are no major paper formatting concerns.

**Quality:**

3

**Strengths And Weaknesses:**

Overall, the paper offers a creative and technically appealing proposal. The mathematical development of the CANN codebooks is careful and correct; the derivations tie neatly to the implementation, demonstrating solid technical quality. The work sits at an interesting and important intersection of machine-learning world models and neuroscientific cognitive map theory, and, if substantiated, could inspire new latent-space designs for long-horizon prediction while providing hypotheses about grid-cell coding. The conceptual novelty is genuine: although VQ-based world models and cognitive-map transformers exist, none combine several CANN lattices with vector quantization in order to achieve the stated architecture-based biological realism and improvements over existing world model implementations. Finally, the prose is readable and the related-work section does a good job of situating the contribution.

Despite this, the paper displays several significant limitations. The model is benchmarked against only a single, hand-rolled baseline, with no modern Dreamer- or RSSM-family comparisons and no statistical uncertainty reported; planning success is illustrated with image strips rather than quantitative metrics. Including further benchmarks could aid in cementing a model such as this as an important bridge between the fields of computational cognitive mapping and world modelling. Important implementation details such as hyper-parameters, compute budget, and codebook ablations are relegated to a one-page appendix, undermining reproducibility. The paper may gain improvements in readability by moving the technically dense description of the CANN architecture to the Methods section. (which, though not unique to this paper, has several parts of its description which are crucial to the understanding of the new model). Several of the figures are confusing, particularly Figure 2. This figure is too dense in its current state to be introduced so early in the text, and should more clearly indicate that the blank rows indicate separate CANNs. The paper would benefit from a few more additions in order to support its claims about biological insight: namely, some discussion of the analog to place cells would be appropriate in a model meant to emulate hippocampal dynamics at a fine enough grain to have grid cell analogs. Additionally, the paper makes a potentially bold stride away from biological realism in its hard-coded actions and should include either further justification of this choice or some discussion of how this could be rectified in future work. The paper also makes a bold claim in its discussion about current world model solutions facing collapse, which should be cited or shown in experimental comparisons against the new model. Finally, the discussion section lapses into a conversational tone that dilutes the scholarly narrative, and there are a few minor spelling and grammatical errors which should be screened for the final version.

---

> ### Author Rebuttal · Authors · 2025-07-28
>
> Thank you for your positive and insightful review. It is rewarding to hear that you found our proposal 'creative and technically appealing' and the mathematical derivations of 'solid technical quality'.  We are glad that you recognized our work has the potential to inspire new latent-space designs, especially for challenging long-horizon prediction tasks. We believe this brain-inspired perspective allows us to introduce a novel and effective architecture to the machine learning community. Your appreciation—acknowledging that our 'conceptual novelty is genuine', the theoretical derivations show 'solid technical quality', and the paper is 'readable'—are greatly encouraging to us.
>
> ---
>
> ### **Responses to Questions**
>
> * We thank the reviewer for this insightful question. It allows us to clarify the conceptual framing of our work and its relationship with computational neuroscience.
>
>   &nbsp;
>
>   To begin, we'd like to outline the distinct functional roles of the two cell types in question. Grid cells are fundamentally associated with path integration \[1]. Their activity is conditioned by self-motion (i.e., they are **bound to action**) and they provide a universal spatial metric that does not globally remap across different environments. In contrast, place cells are crucial for localization by recognizing specific places. Their firing is driven by environmental sensory cues, causing them to **remap** when the environment changes \[2], and they are not directly bound to specific actions.
>
>   &nbsp;
>
>   Given this functional distinction, the design of our model and its core objective naturally align with the principles of grid cells. The main reason for not considering place-like cells is that our model aims to build an **action-conditioned world model**, a task we see as a generalization of the principle underlying grid cells. Specifically, the world model learns to predict a future latent state based on the current state and a given action ($s_{t+1} = s_t+a_t$) as expressed by Eq.(6-7). This process of modeling the environment's transition dynamics is a generalized form of path integration. Consequently, the latent representation our model utilizes is naturally a grid-cell-like code that is bound to the applied action.
>
>   &nbsp;
>
>   The emergence of place cells, on the other hand, is tightly linked to binding specific sensory inputs (e.g., visual landmarks) to locations. Our model is not optimized for this sensory-associative function. By focusing on the generalized transition dynamics, our architecture is designed to factor out environment-specific sensory details in favor of a universal state representation. Therefore, the absence of place cells is confirmatory evidence that our architecture has effectively isolated and replicated the specific neural computational mechanism we target for: a universal, action-based state prediction.
>
>   &nbsp;
>
>   We agree that integrating a place-cell-like mechanism is a valuable direction for future work. As the reviewer suggests, this would sharpen the model's biological grounding and expand its capabilities. For instance, in future work on SLAM tasks where recognizing and relocalizing within previously seen environments is crucial, a place-cell-like module would be essential. This module could help bind the grid-like representations from our current modules to a specific environmental context, providing a mechanism for global remapping. However, this is beyond the scope of our current contribution.
>
> * Regarding your second point, we would like to provide a clear justification for our choice of hard-coded actions. Our approach is grounded in experimental evidence showing that grid-like bumps shift in correlation with an animal’s movements [3]. In our model, this principle translates to the mapping between an action and its corresponding displacement in the latent CANN space. While the scaling factor of this mapping could be learnable, we opted to fix it in our design.
>
>   &nbsp;
>
>   To validate this choice, we conducted an ablation study where the scaling factor was learnable. The results showed a decline in performance compared to using hard-coded actions. We hypothesize that this is because a non-stationary (i.e., constantly changing) latent transition destabilizes the training process. A constantly shifting transition makes it significantly more difficult for the model to learn the mapping between latent states and observations, thus hindering convergence. The outcomes of this ablation are shown in the table below (For FIDp, lower is better. For PSNRp, higher is better).
>
>   |             |   FIDp | PSNRp |
>   | ----------- | ------ | ------ |
>   | GCQ (fixed)   | 43.41 |27.77|
>   | GCQ (learnable) | 47.76 | 24.48|
>
> We hope this explanation and ablation study address your **two main questions**. We will incorporate these clarifications into the final version of the paper to address the ambiguity. Please don’t hesitate to discuss them with us if further questions arise. Now, we turn to your remaining questions.
>
> * We agree with your recommendation to move the continuous attractor derivations from Section 3 to the **Methods** section in the final version for improved clarity and narrative flow.
>
> * Regarding Figure 2, we will revise it to add an inset zoom on the “multi-CANN codebook to decoder” path, to more clearly indicate that the blank rows correspond to separate CANNs. (Apologies that we cannot include the revised figure here due to the rebuttal policy.)
>
> * Thank you for pointing this out. We will add reference [4] to support the statement on “instability issues commonly seen in current world models.” Additionally, we would like to point out that the experiment shown in Figure 4(D) provides empirical evidence, showing that the performance of the VQ method degrades as the prediction horizon increases.
>
> * We will also revise the phrasing in the last paragraph to adopt a more academic tone.
>
> ---
>
> ### **Responses to Weaknesses**
>
> * As you suggested, we have conducted a comparative experiment with a stronger benchmark by replacing the UNet with a Transformer whose architecture is based on TransDreamer [6] (Transformer-SSM is a variant of RSSM). The table below shows the FIDp score as a function of the prediction step, similar to the analysis in Fig. 4D. Lower is better.
>
>   |    Prediction Steps  |  1  | 2   | 4 | 8 |16|32|
>   | ----------- | ------ | ------ | ------ |------ |------ |------ |
>   | VQ+Unet (96M) | 51| 78| 115| 149| 159| 164|
>   | VQ+Transformer (121M) | 61 | 64 | 71 | 79 | 88 | 111 |
>   | GCQ (112M) | 42| 44| 43| 44|44 |44|
>
>   Additionally, we report the planning success rate as a quantitative metric for evaluation(2DMaze dataset).
>
>   |             | 542M   | 307M   | 140M   |
>   | ----------- | ------ | ------ | ------ |
>   | seen goal   | 97/100 | 93/100 | 68/100 |
>   | unseen goal | 91/100 | 88/100 | 65/100 |
>
>   We believe these additional benchmarks strengthen our model's claims. While we were unable to complete the full statistical uncertainty analysis due to time constraints, we commit to including these results and the complete analysis in the final version of the paper.
>
>   We also commit to including more detailed hyper-parameters, compute budget, and codebook ablations in the final appendix. We have submitted our experimental code in the Supplementary Material to ensure reproducibility.
>
> * The remaining points you raised—including relocating technically dense description to the Methods section, providing further discussion on place cells and hard-coded actions, citing instability issues in world models along with empirical support, and adopting more academic language in the discussion—have all been addressed in detail in the **Responses to Questions** section above. We also commit to carefully proofreading the final manuscript to eliminate 'minor spelling and grammatical errors'. We sincerely thank you for your attentive review.
>
> ---
>
> ### **Response to Limitations**
>
> As addressed in **Response to Questions**(point two), we provide a detailed justification for the hard-coding of action sets, along with ablation results comparing to a learnable mapping variant.
>
> As clarified in **Response to Weaknesses**, our work focuses on world modeling, and we included an additional benchmark using a stronger modern baseline (TransDreamer). Our experiments on goal-directed planning and inverse modeling are common downstream tasks for world models [5]. In future work, we plan to incorporate other modules such as place cells and boundary cells to support a broader range of downstream RL tasks and SLAM.
>
> ---
>
> In summary, we sincerely thank you again for your positive assessment and constructive suggestions on our work. We have responded thoroughly to your concerns, especially the two main clarification requests mentioned in your *Questions*. We hope our responses help to increase the perceived biological plausibility and methodological rigor of the work. We commit to implementing all proposed revisions in the final version of the paper. We hope our responses have answered all your questions and please feel free to reach out if any further clarification is needed.
>
> ---
>
> [1] Microstructure of a spatial map in the entorhinal cortex.
>
> [2] Hippocampal remapping and grid realignment in entorhinal cortex.
>
> [3] Path integration and the neural basis of the'cognitive map'.
>
> [4] Facing off world model backbones: Rnns, transformers, and s4.
>
> [5] Learning to act without actions.
>
> [6] TRANSDREAMER: REINFORCEMENT LEARNING WITH TRANSFORMER WORLD MODELS.

---

> > ### Comment · Reviewer_imb7 · 2025-08-05
> >
> > Thank you to the authors for your comprehensive replies to the reviews of myself and others. In light of another look at this paper as well as the adjustments made in the rebuttals to myself and reviewer NYvu, I believe my individual scores were a little too low, and I’m happy to increase my overall score to a 5.

---

> > > ### Author Response · Authors · 2025-08-05
> > >
> > > Thank you very much for your thoughtful reconsideration and for taking the time to re-evaluate our work. We sincerely appreciate your constructive review and are glad that our responses were helpful. Your updated score and comments mean a great deal to us, and we are encouraged by your recognition of our work.

---

### Official Review · Reviewer_rhSR · 2025-07-10

**Clarity:** 4
**Significance:** 4
**Originality:** 4
**Rating:** 6
**Confidence:** 4

**Summary:**

Quantization/discretization are widely used in tokenization for generative models, producing compositional or interpretable token , bottlenecking information flow within a model and many other application. However, most discretization method have their own limitation, such as unnecessary information loss and difficulty of optimization due to non-differentiable steps.

The authors proposed a brain-inspired quantization method that uses attractor which allows conditional generation of codes(concepts), which allows the quantizer to be used as a dynamic/world model to capture trajectory of action-state sequences. Under different actions/condition,
the attractor stabilizes at attractor states.  The authors provided a detailed explanation of CANNs , which provides necessary backgrounds to reviewer unfamiliar to the field ,like me. The inspiration of the method design was drawn from grid cells from animal/human brains and the method creates space similar to bump dynamics in human brain, which the authors interpreted as a cognitive map.

Evaluation of the model was conducted on  Street View dataset, 2DMaze and other 2 datasets, in which use cases method really works like a world model

**Questions:**

I am not an expert of attractors but the attactors do not have to stablize , right? Therefore, the method can be used for a mixture of continuous and discrete tokens?

**Ethical Concerns:**

["NO or VERY MINOR ethics concerns only"]

**Final Justification:**

I looked the authors' replies to my questions , as well as to other reviewers' questions, especially reviewer  imb7 and wNwm. The quality of the work is high and the quality of the reply are decent. Therefore, I like to raise my score and wish this work can help improve discrete tokenization methods used widely in generative model community

**Limitations:**

The main limitation is , the experiments only show adv. of the quantization method in dynamic modelling/world modeling applications. However, there many other cases quantization are needed, where are not discussed/experimented on in this study

**Paper Formatting Concerns:**

No concern

**Quality:**

4

**Strengths And Weaknesses:**

Strength:

1) important quantization contribution that conditioned on action
2) interesting idea to use attractor to generate code
3) amazingly show the potential use as a world model
4) detailed explanation inspiration (equations 1-8)
5) well designed and conducted experiments

Weekness:

1) Though it is nice to draw inspiration from the brain , some parts are "over inspired" such as the cognitive map. Simply use description such as "landscape" or "manifolds" will work

2)The experimental design focus on dynamic/world model like behaviors . If that is the major focus of the author, the tittle and introduction need to pitch that way

---

> ### Author Rebuttal · Authors · 2025-07-26
>
> Thanks for your positive and constructive reviews. We are grateful for your recognition of our work's originality and primary contributions—the quantization method and the concept of attractor-as-code—and for your encouraging comments on that our model has potential to be used as a world model and that our experiments are well designed.
>
> ---
>
> ### **Response to Weakness**
>
> 1.  Thank you for the constructive suggestion. We agree that using more standard terminology improves the paper's clarity. Accordingly, we will revise the manuscript by replacing 'cognitive map' with 'attractor landscape' throughout the text. We believe this change makes the concept more accessible to broader audience, and we appreciate your suggestion on this.
>
> 2.  We agree to highlight the focus of our work is on dynamic/world model. Indeed, a key contribution of our work is replacing the conventional codebook in VQ with attractors (GCQ), thereby directly compressing spatiotemporal information in sequential data. As a result, GCQ can handle all tasks that standard VQ can, such as image, video, speech, action, and multimodal data. In the design of our experiments we chose to demonstrate GCQ’s strengths in the context of world modeling, as it is the most natural application. We appreciate this insightful suggestion and agree that emphasizing this aspect more strongly would benefit readers. We will revise both the title and introduction accordingly in the final version to underscore GCQ’s role as a world model.
>
> ---
>
> ### **Response to Questions**
>
> Thanks for raising this insightful question. Your intuition is absolutely correct, and you’ve highlighted a fascinating aspect of attractor networks. While the term "attractor" implies stability, the structure of that stability is not limited to a single, fixed point. This leads to different types of representations:
>
> - Discrete Attractors: In many classic networks (like Hopfield networks), the system stabilizes into one of several isolated, discrete points in the state space. These are ideal for representing discrete tokens.
>
> - Continuous Attractors: It is also possible for attractors to form continuous structures (e.g., a line or a ring of stable states). These are perfectly suited for encoding continuous-valued tokens.
>
> This leads directly to your main point: yes, our method can be used for a mixture of continuous and discrete tokens, specifically, continuous attractors for continuous-valued tokens and discrete attractors for discrete tokens.
>
> ---
>
> ### **Response to Limitations**
>
> We agree that quantization is needed in many contexts. Thus, our method has broad applicability and potential contributions across a range of domains, including those mentioned in the response to Weakness 2. In this paper, we chose to focus our experiments on dynamic/world model settings where GCQ’s advantages are most direct. We also demonstrate an additional application to image compression in Appendix A. We will add discussions about these potential applications in the revised manuscript, and in future work, we will explore them through extensive experiments.
>
> ---
>
> Once again, we sincerely thank you for your recognition of our contributions, the novelty of our idea, and the rigor of our experimental studies. Your positive evaluation and feedback are invaluable to us. We also appreciate your constructive suggestions on terminology, presentation, and broader applicability. We are committed to incorporating the corresponding revisions in the final version of the paper.

---

> ### Comment · Reviewer_rhSR · 2025-08-04
>
> I appreciate the detailed reply from the author and answered all my questions
>
> I looked the authors' replies to my questions , as well as to other reviewers' questions, especially reviewer  imb7 and wNwm. The quality of the work is high and the quality of the reply are decent. Therefore, I like to raise my score and wish this work can help improve discrete tokenization methods used widely in generative model community

---

> > ### Author Response · Authors · 2025-08-04
> >
> > Thank you very much for your positive and encouraging feedback on our rebuttal. We are delighted to hear that our responses addressed your questions and that you have raised your score for our work.
> >
> > We are particularly grateful for your kind words regarding the high quality of our work, the decency of our reply and potential to contribute to the generative model community. Your specific mention of our responses to reviewers imb7 and wNwm underscores your thorough and careful consideration of our submission, which we deeply appreciate.
> >
> > Thank you again for your time and effort during the review process.

---

### Comment · Area_Chair_FhBL · 2025-08-04

Hi, all.

Thanks for the productive discussion so far. I'd like to remind reviewers who've not checked in that we have only two days left in the discussion period, and I'm sure authors would appreciate the opportunity for some back-and-forth in case there are remaining questions.

Thanks,
AC

---

### Note · Authors · 2025-08-12

We sincerely thank the reviewers for their active engagement and meticulous review. We are glad that all reviewers acknowledged the novelty of our work and felt their concerns were fully addressed in our rebuttal. To better assist the Area Chair's evaluation, we would like to summarize the key contributions of our work, as highlighted by the reviewers, into the following points:

1.  To the best of our knowledge, GCQ is the first model to unify spatial and temporal compression through an action-conditioned quantization process. This enables the direct compression of observation-action sequences, offering an integrated alternative to conventional two-stage world models (reviewer NYvu, rhSR, imb7, wnzM).
2.  GCQ’s spatiotemporal compression yields a cognitive map that supports long-horizon prediction, goal-directed planning, and the derivation of an inverse model. In particular, goal-directed planning becomes computationally simple, as it reduces to finding a sequence of valid bump transitions on the map (reviewer NYvu, rhSR, imb7, wnzM).
3.  GCQ offers insights into the formation of grid cells in the brain, enhancing our understanding of neural representations (reviewer NYvu, rhSR, imb7).

Furthermore, the reviewers recognized the following strengths in our work:

1.  Empirical results show the strong performance of the method on relevant benchmarks, especially for long-horizon prediction and planning (reviewer NYvu, rhSR, imb7, wnzM).
2.  Actions correspond to explicit bump shifts in the attractor network, which yields a cognitive map with an interpretable latent structure conducive to inverse modeling (reviewer NYvu).
3.  The mathematical development of the CANN codebooks is careful and correct, and the derivations tie neatly to the implementation, demonstrating solid technical quality (reviewer rhSR, imb7).
4.  It appears that GCQ is compatible with any encoder-decoder backbone, and we present an ablation study of different backbones that shows the ViT backbone strikes a good trade-off between efficiency and performance (reviewer NYvu).

We are delighted that the reviewers believe all their concerns have been addressed and that all of them ultimately offered their strong support for our work. We are committed to incorporating the reviewers' insightful suggestions, along with the extra experiments and clarifications from our rebuttal, into the final version of the manuscript.

---

### Decision · Program_Chairs · 2025-09-17

**Decision:**

Accept (spotlight)

**Comment:**

This paper introduces Grid-like Code Quantization (GCQ), a method for tokenizing world models. In distinction to typical approaches that uses VQ-VAE and similar models to discretize states in latent space without reference to action, GCQ builds on ideas from grid cells in the neuroscience literature to construction a series of toroidal attractors that encode (world, action) pairs in the algebra of attractor bump positions. Through a series of experiments, the authors show improvements in both planning and long-horizon prediction, as well as FID scores on scene reconstruction. The model works with any encoder-decoder backbone, allowing it to be used as a drop-in replacement in many training pipelines.

Reviewers were enthusiastic about the novelty of the proposed method, though they were mixed about the clarity of presentation. Some reviewers found the experimental results very strong, while others noted they were somewhat limited in scope. Overall, however, the consensus was that this was a well-executed study likely to be of interest to both the neuroscience and machine learning communities.

During rebuttal, authors clarified several conceptual points and proposed readability improvements. They also provided additional results using a vision transformer in place of a U-net backbone, strengthening results. At the end of this period, reviewers showed a strong consensus for acceptance.